# Taxonomic Evaluation of a Bioherbicidal Isolate of *Albifimbria verrucaria*, Formerly *Myrothecium verrucaria*

**DOI:** 10.3390/jof7090694

**Published:** 2021-08-26

**Authors:** Mark A. Weaver, Robert E. Hoagland, Clyde Douglas Boyette, Shawn P. Brown

**Affiliations:** 1USDA-ARS, Biological Control of Pests Research Unit, Stoneville, MS 38776, USA; Doug.boyette@usda.gov; 2USDA-ARS, Crop Production Systems Research Unit, Stoneville, MS 38776, USA; bob.hoagland@usda.gov; 3Department of Biological Sciences, University of Memphis, Memphis, TN 38152, USA; spbrown2@memphis.edu

**Keywords:** *Myrothecium verrucaria*, *Albifimbria verrucaria*, taxonomic classification, bioherbicide

## Abstract

The fungal genus *Myrothecium* was once polyphyletic but a recent reconsideration of the family *Stachybotryaceae* spilt it into several genera. The ex-neotype specimen of the species *Myrothecium verrucaria* is now recognized as *Albifimbria verrucaria*. The well-studied plant pathogen and candidate bioherbicide CABI-IMI 368023, previously identified as *M. verrucaria*, was analyzed morphologically and genetically and found to be most consistently aligned with the other representatives of *A. verrucaria*.

## 1. Introduction

The genus *Myrothecium* was named by Tode in 1790 and has been revised and reconsidered numerous times since then. Tulloch [1] provided a thorough examination of the genus drawing from direct examination of materials from several collections and developed a taxonomic key. However, substantial difficulty was noted in differentiating among species within the genus and even between closely related genera, often relying on admittedly superficial characteristics. Over the years, the members of this genus have attracted attention due to their abundant production of lytic enzymes and the presence of highly unique mycotoxins [2,3,4,5]. While relatively few members of the genus are reported as pathogenic on plants and generally as ‘minor’ or ‘weak’ pathogens, a few isolates are known to be highly virulent on several weedy plants [6,7,8]. One isolate, collected from a diseased sicklepod (*Senna obtusfolia*) in DeSoto County (Mississippi, USA) morphologically identified as *M. verrucaria* and deposited as CABI-IMI 368,023 [9], has been thoroughly investigated as a bioherbicide for some difficult-to-control weeds in aquatic, agronomic and non-agronomic settings [10,11,12,13,14].

DNA sequencing offers a powerful tool in taxonomy and can confirm or dispel long-standing taxonomic understanding. Chen et al. [15] reviewed the existing morphologic descriptions of *Myrothecium* coupled with ITS and EF1-α sequence data, as well as several related genera for which sequence data were publicly available. They concluded that *Myrothecium* was polyphyletic and that the species concept was not well resolved. Working at the same time, Lombard et al. [16] sequenced six genes from nearly one hundred type specimens (ex-type or ex-epitype) and several hundred other isolates from public and private collections. These sequence data were used to revise the family *Stachybotriaceae (Hypocreomycetidae, Sordariomycetes, Pezizomycotina* and *Ascomycota)*. In that examination of *Stachybotriaceae,* the genus name *Albifimbria* (a reference to the white fringe around the sporodochia) was introduced and representatives previously described as *M. verrucaria* received the epithet *A. verrucaria* as the typified representative. In addition, several other former members of *Myrothecium* were placed in *Albifimbira*, including *A. lateralis*, *A*. *terrestris* and *A*. *viridis*.

The recognition of CABI-IMI 368,023 as *M. verrucaria* pre-dated DNA-based taxonomic consideration. Therefore, in the context of the substantial revisions within *Stachybotriaceae* and the importance of this particular isolate, we compared six genes and photomicrographs of CABI-IMI 368,023 with published sequences and images of *Albifimbria* spp. and closely related taxa.

## 2. Materials and Methods

### 2.1. Phenotypic Characterization

CABI-IMI 368,023 was originally isolated from diseased sicklepod (*Senna obtusfolia*) in DeSoto County (Mississippi, USA) and has been maintained at the USDA-ARS in Stoneville, MS, USA. After seven days of growth on potato dextrose agar (PDA) at 28 degrees, photomicrographs were taken of CABI-IMI 368,023 with a Keyence VHX5000 light microscope from 20× to 2500× magnification with integrated digital measurements. Digital images were recorded, and morphology was compared to representative images from Lombard et al. [16].

### 2.2. Genetic Characterization

Genomic DNA was extracted from the fungal isolate CABI-IMI 368,023 with the Zymo Fungal Miniprep Kit (#D6005) from approximately 10^8^ conidia. The genome was sequenced using one Spot-ON Flow Cell (R9) on the Oxford Nanopore Technologies (ONT) MinION platform following ligations using standard nanopore protocols (Ligation Kit SQK-KSK109). Sequencing was conducted over 72 h, and base calling was conducted with the Fast Basecalling implementation by using the innate MinKNOW GUI with a minimum Q-score = 7 for sequence inclusion. This resulted in 8.5 × 10^9^ called bases, 1.3 × 10^6^ sequences, average sequence length = 6154 bp and longest read = 75,816 bp with raw data of N50 = 8438. Since the true identity of this isolate was unknown, we extracted the Internal Transcribed Spacer (ITS) regions within silico PCR within the program Geneious (v.11.1.5) by using the primers detailed in Lombard et al. [16]. A BLASTn query of the *ITS* sequence against GenBank revealed >95% identity with the type specimen of *Albifimbria verrucaria* as well as with *A. lateralis* and *A. virdis*. Since the *ITS* sequence identity alone has limited power to differentiate *Albifimbria,* additional queries were made against 4 *Albifimbira* species and 20 other *Stachybotriaceae* using the *cmdA*, *ITS*, *LSU*, *rpb2*, *tef1* and *tub2* genes, as in Lombard et al. [16]. These gene sequences, from all 19 *Albifimbria* isolates (12 *A. verrucaria*, 3 *A. terrestris*, 3 *A. viridis* and 1 *A. lateralis* strain) used by Lombard et al. [16] as well the sister genera *Dimorphiseta*, *Smaragdiniseta*, *Parvothecium, Inaequalispora*, *Virgatospora*, *Peethambara*, *Septomyrothecium* and *Paramyrothecium roridum* NRRL 2183 (the only closely related species that is entirely sequenced), were concatenated and aligned using MUSCLE (v.3.8.425) as implemented in Geneious (v.11.1.5), and a Neighbor-Joining Tree was generated (Tamura-Nei distance model) by using *Paramyrothecium roridum* as an outgroup and resamples using bootstrapping with 1000 replicates.

## 3. Results and Discussion

Figure 1 presents microscopic observations that support the placement of this isolate within *A. verrucaria*. Measurements of ten conidia yielded an average length of 7.0 µm ± 0.9 and width of 2.7 µm ± 0.5 µm. When grown on PDA, the mycelium is white with a buff reverse. Masses of spores range from very dark olive to nearly black and clustered in moist masses. These features are as described in figures and text by Tulloch [1], with the exception of the lack of fantailed appendages on the conidia.

Generic sequences of the cmdA, ITS, LSU, rpb2, tef1 and tub2 genes from CABI-IMI 368,023 are deposited in GenBank (MZ673262–MZ673264). The type specimen of *Albifimbria verrucaria* was the closest match to CABI-IMI 368,023 for four of the six genes examined (cmdA, LSU, rbpb2 and tub2), with identity of 99.2% to 99.9% between the two isolates (Table 1). The other two genes (ITS and tef1) showed 99.8% and 96.2% identity, respectively, to *A. verrucaria.* The ITS and LSU genes had high homology relative to many isolates of other genera, indicating that they had poor taxonomic resolution. The cmdA and rbp2 genes appeared, in the context of these isolates, to best differentiate *A. verrucaria*. In order to further examine the placement of CABI-IMI 368,023 within this clade of *Stachybotriaceae*, we constructed a phylogenetic tree of likely close relatives, including the *Paramyrothecium roridum* as the outgroup (Figure 2). With the recent reconsiderations of the family, due to emerging genetic resources [15,16], this clade appears to be well-resolved with the monophyletic branches. CABI-IMI 368,023 is present in a division with only other *A. verrucaria* with very low genetic differences separating represented members of species.

## 4. Conclusions

The morphological and genetic observations here support the renaming of CABI-IMI 368,023 as *Albifimbria verrucaria*, consistent with the assignment of other isolates previously recognized as *Myrotheicum verrcuaria*.

Current Name:

*Albifimbria verrucaria* (Alb. and Schwein.) L. Lombard and Crous, in Lombard, Houbraken, Decock, Samson, Meijer, Réblová, Groenewald and Crous, Persoonia 36: 177 (2016).

Synonymy:

*Peziza verrucaria* Alb. and Schwein, Conso. Fungi. (Leipzig): 340 (1805) (also the Basionym).

*Gliocladium fimbriatum* J.C. Gilman and E.V. Abbott, Iowa St. Coll. J. Sci. 1: 304 (1927).

*Metarhizium glutinosum* S.A. Pope, Mycologia 36(4): 346 (1944).

*Myrothecium verrucaria* (Alb. and Schwein.) Ditmar, in Sturm, Deutschl. Fl., 3 Abt. (Pilze Deutschl.) 1(1): 7 (1813).

## Figures and Tables

**Figure 1 jof-07-00694-f001:**
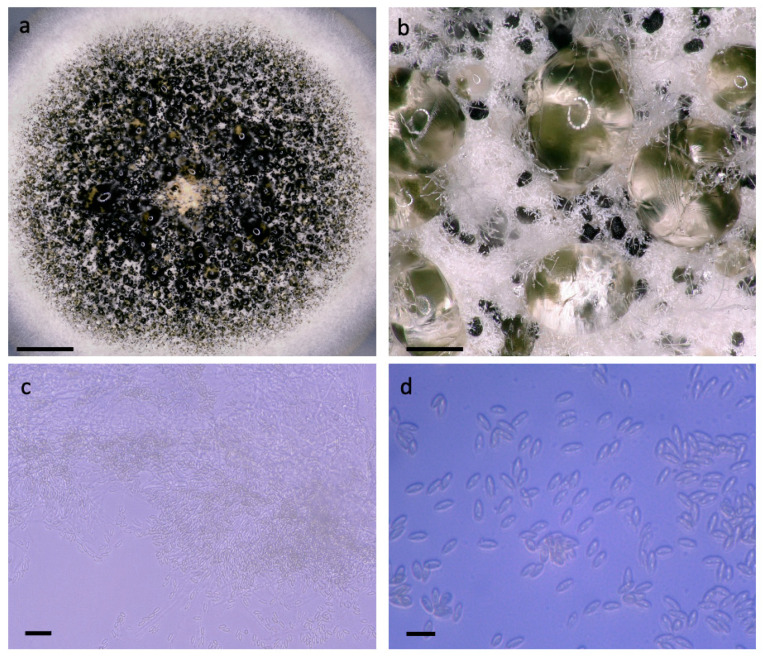
Photomicrographs of CABI-IMI 368023. Images (**a**–**d**) are captured on Keyence VHX5000 light microscope at 20, 200, 1000 and 2500× magnification, respectively. (**a**) Culture morphology with dark region of sporulation surrounded by white mycelial margin, bar = 5 mm. (**b**) Dark spore masses embedded in white arial mycelia and droplets of condensation, bar = 400 µm. (**c**) Closer examination of free conidia and dense hyphal mass, bar = 25 µm. (**d**) Elongated conidia, lacking fantail appendage, bar = 10 µm.

**Figure 2 jof-07-00694-f002:**
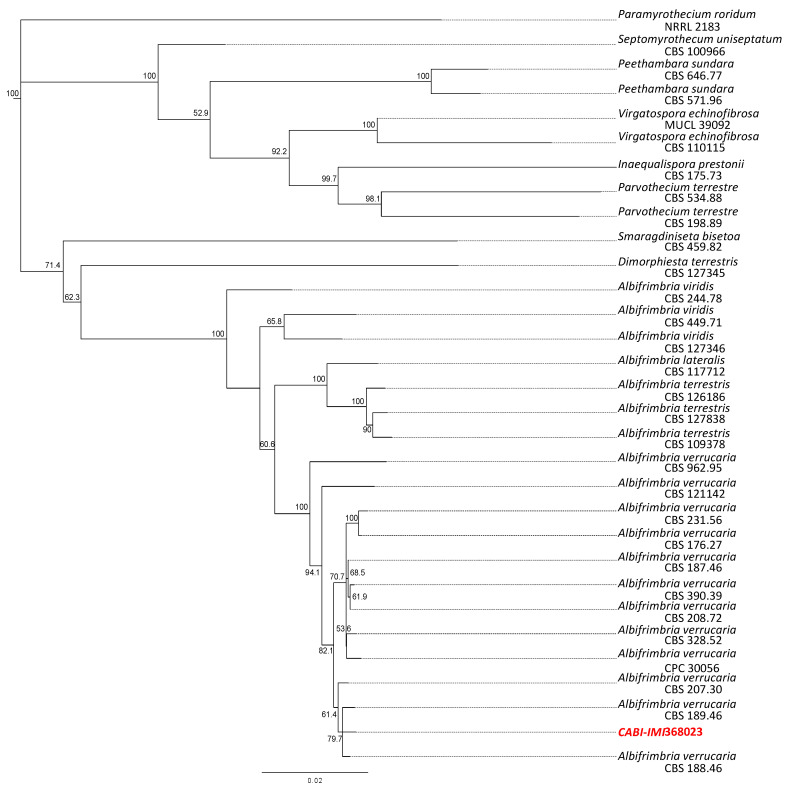
Consensus six-gene concatenated neighbor-joining tree placing the bioherbicidal isolate IMI 368,023 (highlighted in red) within *Albifimbria verrucaria*.

**Table 1 jof-07-00694-t001:** Genetic similarity of *Albifimbria verrucaria* strain CABI-IMI 368,023 to other *Albifimbria* taxa and select *Stachybotriaceae*. Presented are percent identical sequences with the number of exact matches/total length presented parenthetically following BLASTn comparisons for the genes calmodulin (*cmdA*), internal transcribed spacer 5.8S (ITS), 28S RNA large subunit (LSU), RNA polymerase II second subunit (*rpb2*), translation elongation factor 1-alpha (*tef1*) and beta-tubulin (*tub2*). Accession numbers for the gene sequences used are presented in Lombard et al. (2016). Instances of >98% identity are highlighted in grey.

Species and Isolate Number	*cmdA*	ITS	LSU	*rpb2*	*tef1*	*tub2*
Achroiostachys humicola CBS 868.73 *^,†^	73.4% (300/409)	73.6% (521/708)	97.0% (801/826)	76.2% (550/722)	90.1% (127/141)	84.3% (305/362)
*Albifimbria lateralis*CBS 117,712 *^,†^	91.3% (608/666)	99.4% (666/670)	99.3% (820/836)	94.6% (682/721)	89.4% (405/453)	96.7% (352/364)
*A. terrestris* CBS 126,186 *^,†^	91.3% (605/663)	99.4% (666/650)	99.2% (819/826)	93.9% (675/719)	89.2% (405/454)	96.7% (551/363)
*A. verrucaria* NRRL 2003 *^,†,‡^	98.5% (649/659)	99.8% (633/643)	99.9% (825/826)	99.3% (716/721)	96.2% (430/447)	99.2% (356/359)
*A. virdis* CBS 449.71 *^,†^	93.3% (615/659)	99.9% (669/670)	99.8% (824/826)	95.4% (683/716)	88.8% (404/455)	98.3% (353/359)
*Brevistachys variabilis* CBS 141,057 ^†^	79.9% (222/278)	77.2% (319/413)	93.5% (772/826)	72.0% (321/446)	92.6% (137/148)	80.8% (298/369)
*Capitofimbria compacta* CBS 111,739 *^,†,‡^	77.7% (313/402)	75.8% (503/664)	98.6% (815/827)	79.6% (577/725)	97.3% (143/147)	82.7% (296/358)
*Dimorphiseta terrestris* CBS 127,345 *^,†^	75.0% (437/583)	81.9% (550/672)	98.6% (815/827)	85.6% (612/715)	94.4% (135/143)	88.6% (327/369)
*Grandibotrys pseudotheobromae* CBS 136,170 *^,†^	--	74.7% (519/695)	96.7% (771/797)	72.9% (510/700)	91.6% (131/143)	81.7% (294/360)
*Gregatothecium humicola* CBS 205.96 *^,†^	77.8% (312/401)	80.6% (555/689)	99.2% (820/827)	82.7% (596/721)	98.6% (139/141)	84.1% (301/358)
*Inaequalispora prestonii* CBS 175.73 *^,†,‡^	73.4% (315/429)	79.0% (527/667)	97.8% (809/827)	79.6% (575/722)	91.6% (130/142)	80.3% (236/294)
*Kastanostachys aterrima* CBS 101,310 *^,†^	--	--	95.5% (790/827)	72.6% (525/723)	--	--
*Myrothecium inundatum* CBS 275.48 *	74.7% (331/443)	80.7% (453/673)	98.6% (815/827)	--	96.5% (136/141)	84.1% (310/358)
*Myrothecium simplex* CBS 582.93 *	75.5% (320/424)	81.0% (545/673)	98.6% (815/827)	--	97.2% (137/141)	83.8% (300/358)
*Neomyrothecium humicola* CBS 310.96 *^,†^	76.3% (318/417)	82.7% (548/663)	99.2% (820/827)	83.4% (601/721)	98.6% (139/141)	--
*Paramyrothecium cupuliforme* CBS 127,789 *^,†,‡^	78.7% (329/418)	81.0% (554/684)	99.2% (820/827)	81.2% (521/642)	91.5% (129/141)	85.0% (306/360)
*P. roridum* NRRL 2183 ^†,‡^	78.5% (328/418)	94.1% (642/682)	98.9% (818/827)	82.0% (591/721)	89.3% (134/150)	84.9% (304/358)
*P. viridisporim* CBS 873.85 *	78.7% (329/418)	81.1% (553/682)	98.9% (818/827)	82.4% (594/721)	89.8% (132/147)	83.8% (300/358)
*Parvothecium terrestre* CBS 198.89 *^,†^	75.1% (320/426)	79.9% (531/665)	97.9%(810/827)	80.6% (582/722)	93.1% (134/144)	82.4% (305/370)
*Smaragdiniseta bisetosa* CBS 459.82 *^,†,‡^	76.8% (447/582)	82.7% (554/670)	99.2% (819/826)	83.8% (604/721)	93.6% (132/141)	87.1% (317/364)
*Stachybotrys chartarum* CBS 182.80 *	74.6% (309/414)	76.3% (509/667)	96.3% (796/827)	75.8% (547/722)	92.9% (131/141)	81.8% (296/362)
*Striaticonidium cinctum* CBS 932.69 *^,†,‡^	76.0% (319/420)	74.9% (500/668)	97.9% (809/826)	78.9% (569/721)	--	82.3% (307/373)
*Tangerinosporium thalitricola* CBS 317.61 *^,†^	73.3% (431/588)	80.5% (542/673)	97.9% (810/827)	--	--	84.7% (305/360)
*Xenomyrothecium tongaense* CBS 598.80 *^,†,‡^	72.5% (427/589)	79.3% (548/691)	98.9% (818/827)	82.1% (592/721)	97.9% (140/143)	86.6% (310/358)

* Type specimen; ^†^ gen. nov. from Lombard et al. [16]; ^‡^ formerly placed into Myrothecium.

## Data Availability

Gene sequences have been deposited with National Center for Biotechnology Information, GenBank. Accession numbers MZ673262–MZ673264.

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
