# Peer review of "Taxonomic Evaluation of a Bioherbicidal Isolate of Albifimbria verrucaria, Formerly Myrothecium verrucaria"

_jof, 2021, doi:10.3390/jof7090694_

Round 1
Reviewer 1 Report
I am quite perplexed with the relevance of this paper where authors simply confirm ascription to Albifimbria verrucaria of a single isolate previously classified as Myrothecium verrucaria. Actually, all isolates classified in a species moved to a newly established genus can be labeled with the new name unless conversely proven. So that the work done by the authors looks to be somehow trivial. However, the paper is well written and reports sequencing of markers which could be useful in further taxonomic resassessments. Therefore it can be acceptable if the Editor does not consider prejudicial my advice concerning general significance. If so, there are only a few corrections to be done, that is using italics for Latin names (title and lines 85, 95, 98, 100, 103 and 127). Moreover, caption of fig. 1 must be integrated with a description of what is shown in the four pictures.
Reviewer 2 Report
The manuscript is devoted to a narrow item – taxonomic evaluation of a particular fungal strain, and from the title, it is not clear what was the subject of evaluation.
Material and Methods should start with a brief description of the site (habitat, substrate) from which the strain was isolated. In Abstract, the studied strain is defined as "the well-studied plant pathogen and candidate bioherbicide", but it is unclear where it came from and why it had been chosen as a potential bioherbicidal agent.
The measurements of conidia (lines 85-86) are presented in a strange way, with SD written in parentheses, instead of the regular way like the range of length x the range of width.
The title of Table 1 is too long, complicate, and vaguely written. It needs to be revised and corrected.
The section 3 is titled "Results and Discussion", but in fact, the discussion is absent. For example, there is no any explanation on high similarity (>98%) of some gene sequences between the studied strain and the strains belonging to different species and even genus (Table 1).
All species names throughout the text should be written in italic.
The authors should avoid multiple repetition of the same words in a sentence (see attached PDF file).
All other suggestions are inserted in the PDF version of the manuscript which is attached.
